# Stigma Mechanisms and Outcomes among Sub-Saharan African Descendants in Belgium—Contextualizing the HIV Stigma Framework

**DOI:** 10.3390/ijerph18168635

**Published:** 2021-08-16

**Authors:** Lazare Manirankunda, Aletha Wallace, Charles Ddungu, Christiana Nöstlinger

**Affiliations:** 1Institute of Tropical Medicine Antwerp, 2000 Antwerp, Belgium; cddungu@itg.be (C.D.); cnoestlinger@itg.be (C.N.); 2Department of Medical Sociology and Health Sciences, Vrije Universiteit Brussel, 1050 Brussels, Belgium; alethawallace@hotmail.com

**Keywords:** HIV, stigma, discrimination, Sub-Saharan African migrants, people living with HIV, Belgium

## Abstract

HIV-related stigma and discrimination are recognized barriers to HIV prevention, testing and treatment among people of Sub-Saharan African descent (SSA) origin living in Belgium, but insights into HIV related-stigma mechanisms and outcomes are lacking for this population with high HIV prevalence. Guided by Earnshaw and Chaudoir’s stigma framework (2009), we conducted this qualitative study using 10 focus-groups with 76 SSA community members and 20 in-depth interviews with SSA descendants living with HIV to explore specific HIV-stigma mechanisms and outcomes and underlying drivers. Inductive and deductive thematic analysis showed high degrees of stigma among SSA communities driven by fear of HIV acquisition and misconceptions in a migration context, negatively affecting SSA descendants living with HIV. The results allowed for contextualization of the framework: At the community level, prejudices and stereotypes were major stigma mechanisms, while physical distancing, gossips, sexual rejection, violence and increased HIV prevalence emerged as stigma outcomes. Among SSA descendants living with HIV, enacted, anticipated and internalized stigmas were validated as stigma mechanisms, with witnessed stigma as an additional mechanism. Self-isolation, community avoidance and low utilization of non-HIV specialized healthcare were additional outcomes. These results are relevant for tailoring interventions to reduce HIV-related stigma.

## 1. Introduction

People of Sub-Sahara African (SSA) origin living in Belgium are disproportionately affected by HIV similar to other West European countries. With 30% of newly reported HIV cases (of those with known nationalities) in 2019, they constitute the second largest group after men who have sex with men [1], while they account only for 1.6% of population (Communication Myria, 10 July 2017). In addition, high HIV prevalence rates (i.e., 5.9% among women and 4.2% among men living in Antwerp) [2] as well as substantial rates of post-migration HIV acquisition (22.7%) were reported [3]. While this demonstrates high HIV prevention needs, the prevention demand in this population has been low [4] for different reasons, such as low self-perceived HIV risk [5,6], lack of information on health services [7], prioritizing legal issues such as obtaining papers over health, the need to learn a new language, find housing and childcare, building up new social relationships [8] and social control associated with fear of being seen at the HIV clinic [9]. HIV stigma has been acknowledged to have a negative impact on uptake of HIV prevention, testing, access to care and treatment, adherence to treatment and HIV status disclosure [10,11,12].

According to Goffman (1963) [13], stigma refers to “an attribute that is deeply discrediting within a social interaction” (p. 3), attached to people who are then viewed as deviant through breaking the moral order [14]. Stigma has been shown to be replicated by social inequalities and power structures within society [15].

Earlier studies in Belgium showed that SSA communities perceived HIV as a punishment resulting from crossing sexual norms, tightly related to dominant social and religious norms [7]. As a consequence, SSA women living with HIV for instance were shown to conceal their HIV status because they feared stigma and discrimination [16]. However, little is known about the underlying mechanisms of HIV-related stigma in this context as a barrier to accessing services along the HIV prevention and care continuum, and its specific consequences on the community and people living with HIV.

The HIV-Stigma Framework of Earnshaw and Chaudoir [17] has been used to guide stigma research by describing the process of stigmatization from the perspective of people both living with and without HIV. In this model, HIV is a stigma attribute that is perceived differently by people living without by HIV (here we refer to them as “communities”) and people living with HIV, and triggers specific mechanisms and outcomes in each group. Once communities learn about someone living with HIV, certain HIV stigma mechanisms (i.e., psychological responses to learning about someone’s HIV positive status) at the community level include prejudice (i.e., negative emotions and feelings), stereotypes (i.e., group-based beliefs about HIV and people living with HIV) and discrimination (i.e., behavioral expressions and actions of prejudice directed at people living with HIV). These mechanisms lead to specific stigma outcomes, such as social distancing, low HIV testing uptake and stigmatizing policy support.

For people living with HIV, HIV stigma mechanisms after their HIV diagnosis (i.e., psychological responses to their own HIV diagnosis) comprise enacted stigma (i.e., beliefs that they have actually experienced prejudice and discrimination from others), anticipated stigma (i.e., expectations that they will experience prejudice and discrimination from others in the future) and internalized stigma (i.e., absorbing the negative beliefs and feelings associated with HIV about themselves). The related outcomes are negative mental health consequences, low access to social support and increase of HIV symptoms. The outcomes at both levels mutually influence these mechanisms, creating a self-perpetuating cycle.

Although this framework was primarily designed to understand and measure individual processes of stigmatization, here we used it as guidance for exploring HIV-related stigma and its underlying drivers among communities and people living with HIV originating from SSA living in Belgium. In addition, we aimed to contextualize this framework for SSA descendants living in a high-income setting. It has been pointed out that stigma theories should be constantly re-assessed based on newly emerging empirical evidence [18]. This can inform future tailored de-stigmatizing interventions for vulnerable populations within a concentrated HIV epidemic in resource-rich settings and contribute to further theoretical insights.

## 2. Materials and Methods

### 2.1. Study Design

This cross-sectional, multi-site qualitative study used an interpretive ethnographical approach [19]. Focus group discussions (FGDs) were conducted to assess community perceptions [20], triangulated with in-depth interviews (IDIs) to explore personal experiences, attitudes and feelings of SSA descendants living with HIV.

### 2.2. Sampling

We adopted purposive sampling with maximum variation using an iterative approach to select ‘information-rich’ cases [21] and to achieve a sample assumed to reflect the SSA communities living in Flanders based on gender and origin. Participants were women and men aged 18 years or above, originating from any SSA country independent of their current nationality, French- or English-speaking, and currently living in Belgium. Our sampling approach combined theoretical data saturation with a priori thematic saturation, which is related to the degree to which identified codes or themes—guided by the HIV stigma framework—were exemplified in the data. The recruitment of participants for all FGD and IDs stopped when thematic saturation was achieved.

### 2.3. Recruitment and Informed Consent

Only SSA descendants who had lived longer than two years in Europe were eligible to participate in the FGDs allowing for sufficient experiences of living in migrant communities. FGD participants were recruited through local socio-cultural organizations of SSA communities in Flanders (the Northern Region of Belgium). Community leaders acted as gatekeepers in a variety of settings, such as social gatherings, churches, cafés, hair salons and African shops.

People living with HIV were invited to participate in an IDI irrespective of the number of years of living with HIV or the length of stay in Belgium, assuming that experiences of living with HIV start with the reception of an HIV test result. IDI participants were recruited at the three largest HIV Reference Clinics (HRC) in Flanders (the Institute of Tropical Medicine, the University Hospital of Ghent, and the University Hospital of Brussels) and from an Antwerp-based peer support group of SSA people living with HIV. To account for potentially triggering traumatizing experiences through interviewing, a referral to psychosocial counselors for immediate support was ensured.

Before partaking in the study, potential FGD or IDIs participants were informed about the study objectives, the consent procedures and the audio-recording. All study participants provided written informed consent before data collection.

### 2.4. Data Collection

FGDs and IDIs were conducted by the two first authors (A.W. and L.M.) using a pre-tested topic guide by the Stigma Framework in English or French. Participants’ socio-demographic characteristics were assessed using a short self-reported questionnaire. All data were collected between December 2015 and June 2017.

The FGD topic guide assessed community participants’ perceptions towards HIV, expression of HIV-stigma at the community level and its perceived impact on the lives of people living with HIV and within the communities. FGDs were held in places familiar to community members. FGDs were homogeneously organized by language, gender and age (young men or women aged between 18 and 24 years) to allow for group dynamics facilitating natural and free discussions. A total of 10 FGDs were organized. On average, FGDs lasted about 90 min.

The IDI topic guide explored how participants experienced their HIV diagnosis, HIV disclosure, coping strategies and support received, the meaning and impact of living with HIV, and of HIV-related stigma and discrimination. Interviews took place in confidential and safe locations chosen by the participants, either HIV clinics or participants’ homes. Twenty IDIs were conducted with about 75 min per IDI on average.

### 2.5. Data Analysis

All FGDs and IDs audio-tapes were transcribed verbatim. Quality checks were carried out on 23% of the pseudonymized transcripts. French transcripts were translated into English and cross-checked by the first author (L.M.) who is bilingual. The first two authors (L.M., A.W.) performed a thematic analysis [21,22] applying both inductive and deductive approaches.

Data analysis was performed with NVIVO 10 (QSR International Pty Ltd., Burlington, MA, USA, released in 2012) by the first two authors (A.W. and M.L.). In the first step, the two researchers familiarized with the data performed thematic analysis [23] by applying deductive principles. They developed a codebook corresponding to stigma mechanisms and outcomes of the HIV stigma framework and extracted the corresponding mechanisms and outcomes that emerged from the data to validate or adapt the stigma framework for the context living in Belgium. The researchers compared the categories and kept elements where consensus was obtained.

In the second step, we applied an inductive approach by building and adding new categories as they emerged from the data, under the mechanisms and outcomes. Authors constantly refined the categories to answer the research objectives.

In the third step, data from FGDs and IDIs were triangulated to increase the results’ trustworthiness and validity of the results [21,22].

The final themes were mapped onto the guiding framework resulting in its contextualization. Dissonant codes on all steps were discussed with the research team to reach consensus.

In the fourth step, a series of feedback moments was organized presenting the results to community members during three community meetings (one for young people, two for adults, each meeting being with mixed genders). With these meetings, we sought participants’ views on the quality and accuracy of the results to increase data validity.

### 2.6. Ethical Considerations

The study received the approval of the Institute of Tropical Medicine’s Institutional Review Board (Reference nr. ITG 1053/15)**,** and the Ethical Committees of the University Hospitals of Antwerp (Reference nr. 15/50/549), Brussels (Reference nr. 2015/352) and Ghent (Reference nr. 2016/1367).

## 3. Results

### 3.1. Participants’ Sociodemographic Characteristics

#### 3.1.1. FGD Participants’ Characteristics

Ten FGDs were conducted as follows. Four FGDs were conducted with English and six with French-speaking participants. Four groups consisted of adult men, four groups of adult women, and two groups of young women and men aged under 25 years, respectively.

Table 1 shows the demographic characteristics of *n* = 76 persons (51.3% of men, 48.7%) participants. Most participants were middle-aged and identified as single, Christians, highly educated and settled residents with low employment rates. Sixteen percent of the participant did not have any health insurance, which in the Belgian health care system can serve as a proxy for being an irregular migrant. Participants originated from 12 SSA countries (Burundi (*n* = 8), Burkina Faso (*n* = 2), Cameroon (*n* = 9), DR Congo (*n* = 18), Ethiopia (*n* = 2), Kenya (*n* = 2), Ivory-Coast (*n* = 1), Ghana (*n* = 9), Nigeria (*n* = 10), Rwanda (*n* = 11), Sierra Leone (*n* = 4), Togo (*n*= 1)).

#### 3.1.2. IDI Participants’ Characteristics

Interviewees’ socio-demographic and medical characteristics are detailed in Table 1 and Table 2, respectively. Participants originated from nine countries: DR Congo (*n* = 5), Rwanda (*n* = 3), Cameroon (*n* = 3), Burundi (*n* = 2), Kenya (*n* = 2), Togo (*n* = 2), Liberia (*n* = 1), Guinea Conakry (*n* = 1), Angola (*n* = 1).

They reported being single, Christians and with low income despite being highly educated. All participants were on medication including six participants who did not have health insurance at the time of the study.

### 3.2. HIV-Stigma Mechanisms among SSA Communities

#### 3.2.1. Prejudices

*Irrational beliefs.* FGD participants reported that many people in SSA communities had pre-conceived judgments about HIV and people living with HIV, rooted in information received in the pre-HIV treatment era. For instance, many believed that people living with HIV would spread HIV through physical and social contacts, and through things they had used or touched. Additionally, many believed that HIV would quickly develop into AIDS with physical degradation leading to death. Picturing HIV as a fatal disease fueled fear of HIV and hence people avoided people living with HIV intentionally, or took irrational precautions to avoid HIV infection.


*“When my friend gets a visit of his friend living with HIV…, they are really funny together. But my friend will clean the toilet more carefully after use, wash carefully things he touches and utensils he has used.”*
(Adult woman, FGD 10)

While FGD participants felt that the described fears were irrational, they described how these fears contributed to hostility against people living with HIV.

#### 3.2.2. Stereotypes about SSA People Living with HIV

*Physical signs of HIV.* In particular, participants with little access to HIV-related information believed that people living with HIV would still develop symptoms like skin disease, diarrhea, weight-loss and changes of hair and nails. However, participants who knew people living with HIV living in Belgium observed a change in such perceptions:


*“…sometimes I will think this guy will die very soon, you know. The kind of thought we have… But still I have seen him living over ten years now and he is just doing fine.”*
(Adult man, FGD 1)

*Sexual stereotyping/Promiscuity.* A frequently emerging theme was promiscuity, believing that people’s past or present immoral sexual behavior was responsible for getting HIV.


*“They condemn immediately. They say the person is not serious. They see nothing else. They crucify him/her immediately: he/she must be a sexual vagrant.”*
(Adult woman, FGD 7)

Those who believed that HIV infection resulted from transgressing sexual norms saw HIV as a punishment for promiscuity. Transgressing sexual norms was perceived as more serious for women than for men. For instance, women living with HIV in the community were often labelled as prostitutes. Likewise, in HIV-discordant couples, usually the wife was blamed for bringing in HIV from the outside.

*Depicting people living with HIV as worthless.* Some FGD participants believed that people living with HIV were dirty and useless. This belief was reinforced by the perception of HIV as a fatal disease, and subsequently by HIV infection as “a death sentence”. In addition, some SSA descendants interpreted HIV infection as a curse, to have been afflicted upon people by witchcraft or perceived it as a punishment from the ancestors or God.

*Intentional Transmission of HIV.* Some community members shared stories that certain people living with HIV would intentionally pass on HIV as revenge or by mere wickedness. Terms like ‘criminals’, ‘poison’, ‘killers’ or ‘bombs’ were used to describe this behaviour and to stress the perceived danger.

### 3.3. HIV-Stigma Outcomes among SSA Communities

#### 3.3.1. Physical and Social Distancing

Many participants acknowledged that community members distances themselves from people living with HIV because of their fear of contracting HIV. Therefore, once it was believed that someone was living with HIV, all forms of interactions would stop quickly: No more hugging, shaking hands or kissing.


*“You no longer have the same charisma you had before. It amuses you somehow, you try to avoid the person. It is a psychological matter.”*
(Adult man, FGD 5)

FGD participants recognized the individual impact of such rejection among people living with HIV including self-isolation, withdrawal, depression and eventually suicide.

#### 3.3.2. Avoidance of HIV Testing

The fear of perceived psychosocial consequences of an HIV diagnosis emerged as the main reason that many community members did not get tested for HIV.


*“By the moment you dare [to test] and you have a doubt about yourself, … your whole attitude, physically, spiritually, mentally would change. That’s the greatest thing people try to avoid. … People feel that the outcome of the testing will be bad.”*
(Adult man, FGD 1)

#### 3.3.3. Increased HIV Prevalence

Most of the participants agreed that stigma contributed to the high HIV prevalence in African communities. Ongoing HIV transmission was attributed to unprotected sex, which was seen as indirectly facilitated by non-disclosure of HIV status and denial of HIV diagnosis. A few participants attributed community HV prevalence to intentional HIV transmission and also held beliefs of conspiracy theories, i.e., that HIV was introduced on purpose by white people to harm Africans.

#### 3.3.4. Support of Restrictive Policies

In particular, those who felt that people living with HIV should be obliged to disclose their HIV status to sexual partners were in favor of restrictive prevention policies, like regular mandatory HIV testing. However, the majority of participants considered such measures to be difficult to implement and would further increase discrimination of people living with HIV.

#### 3.3.5. Forms of Discrimination

*Spreading gossips.* All participants felt that people living with HIV were gossiped about too much, which was seen as harmful for them.

Participants described that gossiping usually started when people living with HIV had disclosed their HIV status to others, e.g., friends or when a conflict between partners emerged. Several circumstances were acknowledged to evoke gossiping as well: The presence of perceived AIDS-related symptoms, being seen at an HIV clinic or a suspected AIDS-related death of a partner.


*“Then the person*
*who breaks the secret will say “I only tell you, don’t share this information, this is how the news circulate.”*
(Adult men, FGD 5)

Participants agreed that once gossiping had started, it was difficult to stop. Such news was said to quickly spread within community networks and finally reach people’s home country. In addition, people suspected to be living with HIV were suspiciously observed, thereby perpetuating the gossips.


*“We have so many [people living with HIV*
*] in this city and at so many occasions people talk about them. The gossip about them everywhere at social gatherings, cafés, and even in the church.”*
(Adult women, FGD 4)

Participants, however, valued the positive aspect of gossip as they felt it was intended to protect others against HIV through social contact or sexual contact. Yet, many FGD participants condemned the gossip because it resulted into harming those living with HIV by isolating them, tarnishing their reputation and, as a consequence, damaging their mental health.


*“We Africans are mean. We will kill through gossip… When people say that someone has AIDS, they will reject the person.”*
(Adult women, FGD 10)

In this regard, a particular method of ‘gossiping’ about people suspected to be HIV positive occurred in one community using social media:


*“…, they made a list of men and women and published it on social media as a sort of alert: here is a list of people from our community who are ill [of HIV/AIDS]… Striking is that community members believed that the names were given by the HIV clinics.”*
(Adult woman, FGD 8)

Some IDIs participants reported that even people living with HIV would gossip about others living with HIV, hoping that this may conceal their own HIV-status and thus they could escape the stigma.

Several life domains emerged from the data as affected by gossips: Social interaction, dating and sexual life, low utilization of social support services and denial of the HIV status. Because of social rejection, participants felt that the majority of people living with HIV would definitely conceal their status.


*“A man will never tell you, of course. He feels his whole community [which] is very small [will know]. If they know his whole family life is messed up.”*
(Adult man, FGD 1)

Sexual life was believed to be affected in two ways: Either people living with HIV would abandon sexual intimacy or conceal their HIV status and practice unprotected sex.


*“What is difficult is to get a partner when you live with HIV. Because if you know that you are ill, you do not want to infect others.”*
(Woman, living 2 years with HIV)

*Gender-based violence.* Participants reported that within couples, usually when men were the first to receive an HIV diagnosis, they would not tell their partners. If women were the ones to be HIV positive, men would blame them and even become verbally or physically violent.

*Sexual discrimination.* Sexual discrimination against people living with HIV was acknowledged as a common phenomenon among SSA descendants. It was perceived as a protection strategy for the potentially uninfected partner. Friends or family members reportedly discouraged any sexual relationships with HIV-positive people.


*“I have actually observed that when people know that someone is infected with the disease… they are scared to establish intimacy with you and there is sexual discrimination.”*
(Adult male, FGD 2)

### 3.4. HIV Stigma Mechanisms among SSA People Living with HIV

#### 3.4.1. Receiving an HIV Diagnosis

The interviewees vividly described how their HIV diagnosis came as a psychological shock, many reacted with disbelief, doubted the test result and feared facing death soon.


*“It was hell. It was like the world has ended. … I was not even thinking about what people will say. I knew that I would die. I even wanted to go back to Africa. Yeah, better to go and die there.”*
(Man, living 13 years with HIV)

In addition, many feared gossip in their community and eventual rejection in case of HIV disclosure. Some participants who were diagnosed during their asylum procedure were additionally afraid of their asylum request being rejected and subsequent deportation to their homelands.

#### 3.4.2. Enacted Stigma

*Gossips and social rejection.* Many participants reported social rejection as a common experience. Some participants who had confided their secret about being HIV positive to people they trusted experienced rejection and gossip. They felt hurt and betrayed. Even those who disclosed their status to only trusted persons experienced gossiping and social rejection based on breaches of confidentiality and rumors:


*“In the beginning the lady and myself we were very good friends, she hurt me... she could not keep a secret and she kept on telling friends and friends… she threw everything I have away… at last she just had to get me out from her house.”*
(Man, living 3 years with HIV)

As a consequence, people living with HIV are isolated from the small community they rely on, or they isolate themselves and experience loneliness, which has a long-term effect on mental health.

*Violence.* Violence emerged as another relevant theme. Shared experiences by IDI participants included verbal abuse, psychological and physical violence, mainly intra-couple violence:


*“He (husband) was calling me mad. He was changing into a madness, sometimes he beat me.”*
(Woman1, living 12 years with HIV)

#### 3.4.3. Anticipated Stigma

All IDI participants reported being afraid of people’s reactions once their HIV status was revealed. Gossip, unwelcoming gestures, looks and mere rejection were mentioned by all interviewees. This fear stemmed either from their own experiences or from witnessed stigma in Belgium and abroad.

*Concealment of HIV status.* Many participants decided not to disclose their HIV status to anybody (except their health care providers) to protect themselves against harsh reactions or to protect significant others from enacted stigma. However, concealing their HIV status came under social pressure as people living with HIV avoided closeness. Such conflicting motivations were perceived as painful, as demonstrated by the following quote:


*“My family always tells me: ‘Why do you remain single, you work, you need to find a girlfriend. I do not want*
*to explain to them. If I find a girlfriend, I will have to tell her that I’m like that and she will leave immediately. And not only will she leave me, but she will start to talk to everyone.”*
(Man, living 10 years with HIV)

*Concealment of HIV treatment and clinic visits.* The anticipated stigma challenged adherence to HIV medication for some IDI participants, particularly when they travelled or had visitors.


*“When people come to see you, you have to hide your medication, put your papers away; everything is coded now. Sometimes in my diary, I use fake names for my appointments. I am afraid that someone will discover something and that it will be serious.”*
(Woman2, living 12 years with HIV)

Disclosing their HIV status to at least one person, such as the partner or a close friend, was acknowledged as an effective adherence support. Some patients reported that the medical follow-up visits were critical moments since they avoided meeting community members at all costs. Therefore, they preferred giving other reasons as explanations, such as getting vaccinations.

#### 3.4.4. Internalized Stigma

Most participants reported feelings such as embarrassment, shame and sadness attached to their diagnosis and initial coping with HIV. Those who had acquired HIV recently were often struggling with internalized stigma, while those living longer with HIV described positive developments in their coping styles due to available medical and psychosocial support.

#### 3.4.5. Witnessed Stigma

All participants had observed negative reactions of community members against people living with HIV such as gossip, hostile body language and rejection. Having witnessed stigma and discrimination promoted secrecy.


*“One day a friend told me: « I cannot greet someone who has AIDS ». He said it just like that. And I have this disease…I can say hello but I cannot shake his hand! …, it’s dangerous for me. When I came home I felt sad because we see each other regularly. If she knew, things would be finished between us. To avoid that, you keep your secret, you don’t tell.”*
(Woman, living 3 years with HIV)

### 3.5. HIV Stigma Outcomes among SSA People Living with HIV

#### 3.5.1. Perceived Health Status and HIV Treatment

All IDI participants reported to be in good health and acknowledged the role of effective HIV treatment in sustaining their health. Good health was a strong motivation to adhere to their medication, as they felt that their apparent physical well-being protected them against community stigma.

#### 3.5.2. Mental Distress

Almost all interviewees acknowledged that HIV non-disclosure would lead to self-isolation. Furthermore, constantly managing concealment of their HIV status in anticipation of communities’ reactions provoked chronic stress and depressive thoughts.


*“It brings you down, even more depressed because you think that people you consider to be your family, your friends should support you, instead, they reject you.”*
(Woman, living 14 years with HIV)

#### 3.5.3. Non-Utilization of Peer-Support

Participants generally acknowledged the benefits of peer support, like sharing mutual experiences and breaking the isolation, even those who were not members of such a group. However, fears to meet community members in such groups prevented them from joining available peer-support because they suspect them of gossiping in the community or in their country of origin.

#### 3.5.4. Self-Isolation and Avoidance of African Communities

Many participants described self-isolation as an initial and temporary stage in coping with HIV. For some, social withdrawal was a deliberate reaction to protect themselves against stigma; others chose to socialize with communities different from their own.


*“My social contact with the African community is limited; most of my friends are white. The few people that know now, they treat me as me, they don’t even think about it when they interact with me. And these are all white people I am talking about now.”*
(Man, living 6 years with HIV)

#### 3.5.5. Utilization of Healthcare

Anticipated stigma played a big role in the impeding uptake of healthcare. In particular, utilizing non-HIV specialized care such as primary care and dental care were perceived as challenging due to disclosure difficulties and its consequences. This was based both on stories shared by others and own experiences. For instance, a dentist had told an HIV-positive woman to wait until all other patients were treated, thereby publicly disclosing her HIV status in the waiting room. Such discriminatory incidents discouraged participants to seek care.

### 3.6. Contextualization of HIV Stigma Framework for SSA Communities

Based on the described stigma mechanisms and outcomes, we contextualized the HIV stigma framework (see Figure 1) by first adding three main drivers of stigma grounded in the data including intersecting forms of stigma and discrimination. Secondly, the original mechanisms and outcomes were adapted based on the local context.

#### 3.6.1. Fear of HIV and People Living with HIV

FGD and IDI narratives demonstrated underlying fears of HIV, of casual HIV infection and fear of finally dying of HIV as important drivers of stigma. Fear was often fueled by one’s own or others’ memories of experiences from home countries when people were falling sick and dying quickly. In addition, prevention campaigns they remembered had presented HIV mainly as a killer disease without an available cure. This image was retained, and the fear was further projected onto people living with HIV in the current environment.

#### 3.6.2. Deficit in HIV-Related Information

The lack of information on current developments in the HIV field emerged as an additional driver of stigma. While FGD and IDI participants acknowledged that information about HIV was available in the community, they felt that some people were not ready to inform themselves, or would not believe in current information, for instance about the effects of anti-retroviral treatment for prevention and of undetectable viral loads on HIV transmission. In particular, first-generation migrants of both genders and newcomers who were still strongly attached to their cultural values were described as lacking correct information.

#### 3.6.3. Multiple Forms of Stigma and Migration-Related Stressors

While IDI participants were generally grateful for living in a country with accessible HIV treatment, they also reported suffering from migration-related stressors such as language barriers, stress and anxieties related to the asylum request process and, particularly when refugee status had been denied, lack of (adequate) employment opportunities and perceived racism.

Language barriers created misunderstandings and frustrations, resulting in perceptions of not being welcome and concerns for the future, particularly because speaking the country’s language was seen as a prerequisite for obtaining work.

Moreover, the lack of employment or job opportunities much lower than the obtained educational level affected participants’ self-esteem and gave room for ruminating about their HIV status, which together with the financial vulnerability, aggravated mental health.


*“They should—yeah—allow people living with HIV to work. Like now it is a big hell for me because I’m home [don’t have a job]. Yeah it’s really a big problem for me. But when I’m busy, I don’t think. I don’t think.”*
(Man, living 13 years with HIV)

Candidates to refugee status whose application was denied experienced the denial as a personal tragedy, as illustrated in this quote:


*“If you want to stay you need a residential permit and if you are denied one, you have a double tragedy. You have a lot of stress and the disease can come up very fast because you are not able to take good care of yourself. You’ll be thinking about the procedure of the papers and whatever. Your health will not be the same as before.”*
(Man, living 12 years with HIV)

Other negative experiences were mentioned, e.g., rejection by neighbors, workplace colleagues or statements by political parties such as *“African refugees are a threat to public health by bringing in HIV and other diseases…”.* Such intersecting forms of stigma were found to exacerbate the perceived discrimination, reduce self-esteem and made it difficult for respondents to disentangle different stigmata.

Further to the drivers, only two stigma mechanisms emerged at the community level: Prejudices and stereotypes. Contrary to the original model, we shifted discrimination to outcomes, since its behavioral components (i.e., gossips, violence and sexual discrimination) were mentioned as enacted stigma among people living with HIV. These two HIV mechanisms led to five specific stigma outcomes including social and physical distancing, avoidance of HIV testing uptake, increased HIV prevalence, restrictive policy support and specific forms of discrimination.

On the level of people living with HIV, four main stigma mechanisms were identified (i.e., enacted, anticipated, internalized and witnessed stigma). Other stressors were not related specifically to HIV, but to migration circumstances that also impacted health and well-being, hindering the use of social (peer) support and the uptake of non-HIV specialized healthcare services. They also led to social withdrawal and avoidance of one’s own community networks. These combined stigma outcomes continuously activated HIV stigma mechanisms. In turn, HIV stigma outcomes among people living with HIV, especially self-isolation and avoidance of own communities, fed HIV stigma mechanisms of community members.

## 4. Discussion

This first explorative community study on HIV-related stigma among people of SSA descent living in Belgium documented strong HIV-related stigma and discrimination, its drivers, mechanisms and outcomes. It largely confirms the original stigma framework, and contextualized both mechanisms and outcomes within a migration context. As shown elsewhere [24,25], the HIV stigma framework allows one to visualize the complexity of relationships between drivers, stigma mechanisms and outcomes among communities and people living with HIV. Our findings show that intersecting forms of stigmata impact on their overall health and well-being.

### 4.1. Stigma at the Level of SSA Communities

Our results showed that the fear of people living with HIV is the predominant stigma prejudice among SSA communities, acting at the same time as a driver both fueling and reinforcing the stigmatization process, as noted by Stangl et al. [26] and in their global health and stigma discrimination framework.

Fear is fueled by the strong belief that people living with HIV are contagious through physical and social contacts, while rational knowledge says otherwise. These results corroborate previous study results from the UK [27], the Netherlands [28,29] and the USA [30,31]. More recent research has shown that people react with implicit prejudice against people living with HIV, as expressed through body language, create unfavorable community settings for people living with HIV. Increasing levels of community implicit HIV prejudice were shown to be associated with increasing levels of psychological distress among community residents with HIV [32]. 

In our context, gossiping was revealed as a significant factor in re-iterating existing stereotypes about people living with HIV fueled by fear, particularly among those who are not correctly informed. Given the dense social fabric in the small SSA diaspora [33], gossiping can have detrimental effects on community members, reducing their social capital by increasing anticipated and internalized stigma mechanisms, which in turn increase self-exclusion, fear of being known as living with HIV, impact mental health [34] and decrease the use of social support services [35]. Research in South Africa showed how gossiping generated resistance to medical efforts to deal with HIV [36]. Anticipating the consequences of being known as someone to be living with HIV, communities will avoid HIV testing and therefore contribute to undiagnosed HIV [37], which perpetuates HIV infections and late diagnosis described in these communities [1]. However, gossiping by using social media for blaming or bullying people of being people living with HIV has not been reported elsewhere among SSA community.

The six HIV stigma outcomes found at the community level are in line with other studies showing that SSA community members reject people living with HIV [29,38] resulting in avoiding intimacy, break-up of (sexual) relationships and verbal or physical violence [27].

Community members with access to HIV-related information were aware of the denial of HIV at the community level, the lack of disclosure of people living with HIV and the impact on community HIV prevalence. Stigma as a major barrier to HIV prevention demand and uptake of HIV testing had been described in several previous studies [11,39,40]. While recent additions to prevention such as U = U (undetectable = untransmittable) campaigns could be powerful tools to reduce sexual HIV transmission [41] and dismantle stigma [42], community members and people living with HIV were not sufficiently aware of this development. It is worrisome that some participants supported mandatory HIV testing and criminalization as policies to curb the HIV epidemic, since it has been shown that such approaches would rather reinforce the marginalization of vulnerable groups [43,44].

### 4.2. Stigma among SSA People Living with HIV

Our findings revealed four HIV-stigma mechanisms affecting people living with HIV: Enacted, anticipated, internalized and witnessed stigma, a new theme compared to the original framework. Anticipated stigma was a common and strong mechanism, linked to witnessed stigma shaped by accounts from African home countries. People living with HIV constantly expected that their HIV status could be revealed, and they feared the resulting gossiping [16,30,45]. This provoked continuous distress and led to self-withdrawal from their communities [38,44], cutting themselves off from needed social support [46,47].

Internalized stigma was less pronounced in our sample compared with other studies [41,42]. This can probably be explained by positive coping strategies that the study participants may have adopted over years of receiving effective treatment and available psychosocial support [48,49]. Participants with recent HIV infections were still ashamed about their status and prone to self-stigma and self-isolation.

Concerning HIV-stigma outcomes, findings on mental health and the low use of existing social support are similar to those of the reference framework. Additional stigma outcomes found in our study are self-isolation, partial or no HIV disclosure [50] and avoidance of the community [28,29] and non-HIV specialized health care services. Participants in our study used non-disclosure as their main strategy to escape stigma and discrimination. For many people living with HIV, disclosure remains a big challenge as it has been documented in Sweden [45], Denmark [51], England [52] and Switzerland [53].

Overall, migration context may play an important role in driving stigma: Since African communities are part of an ethnic minority group, perceived stigma may be intertwined with racism, and for women living with HIV, also with gender issues [54]. Existing stereotypes about migrants within the current political discourse on migration in Europe [55,56] may contribute to perceiving multi-layered stigma and discrimination, as confirmed by previous studies that demonstrate that key populations among people living with HIV feel “doubly guilty” [57]. This may explain why stigma and discrimination against people living with HIV persist even though HIV has changed from a fatal to a chronic disease [52].

### 4.3. Strengths and Limitations

These findings must be interpreted against its strengths and limitations. We used a widely validated framework to gain theoretical insights, and sound qualitative methods to enhance the study’s trustworthiness through data source triangulation [58]. To ensure data validity, we took several steps: First, we used multiple informants, using purposively selected samples (i.e., in terms of gender, age, ethnic origin, education level, length of living with or without HIV in Europe). This created a more accurate understanding of the issues at stake and allowed for data source triangulation. Second, all data were collected using non-directive questioning techniques adopted by trained qualitative researchers, both in the FGDs as in the IDIs. This likely yields accurate information. Third, we obtained feedback from the communities on the research results, which confirmed the findings’ accuracy, and we obtained input for their application to future stigma reduction interventions.

We acknowledge a potential recruitment bias. We may have included participants who were more open to discuss HIV, thus potentially underreporting stigma. Further studies using mixed or quantitative methods should determine the prevalence of stigma and discrimination in the general SSA population, allowing for monitoring of its change over time.

## 5. Conclusions

This qualitative study demonstrates a high degree of community-level stigma characterized by irrational beliefs regarding HIV and people living with HIV. Many stereotypes trigger social and physical distancing of and permanent gossiping about people living with HIV. Intersecting HIV- and migration-related stigmata impact negatively on the social interactions and mental health of people living with HIV. The contextualized HIV Stigma Framework shows, in detail, how mutual interactions between outcomes and mechanisms at both the community level and among people living with HIV occur, fueled by multiple underlying drivers. These theoretical insights help to identify priorities for stigma reduction in the current context.

For effective stigma reduction, we recommend developing multi-level interventions that target HIV stigma mechanisms and drivers. The emotional reactions to people living with HIV such as fear and the related fear of death could be mitigated by tailored evidence-based interventions promoting interactions between community members and people living with HIV developed through participatory approaches. People living with HIV should be involved in primary HIV prevention to provide the role models for healthy living with HIV.

To support people living with HIV in coping with community stigma, tailored empowerment interventions should be implemented among newly diagnosed people, and peer support on individual and group levels should be strengthened to create safe spaces where they can enjoy confidentiality and an open culture benefitting their health and mental well-being.

To target enacted stigma and discrimination at the health-care level, tailored stigma reduction interventions targeting primary health care professionals should be developed in co-creation with affected communities and people living with HIV.

## Figures and Tables

**Figure 1 ijerph-18-08635-f001:**
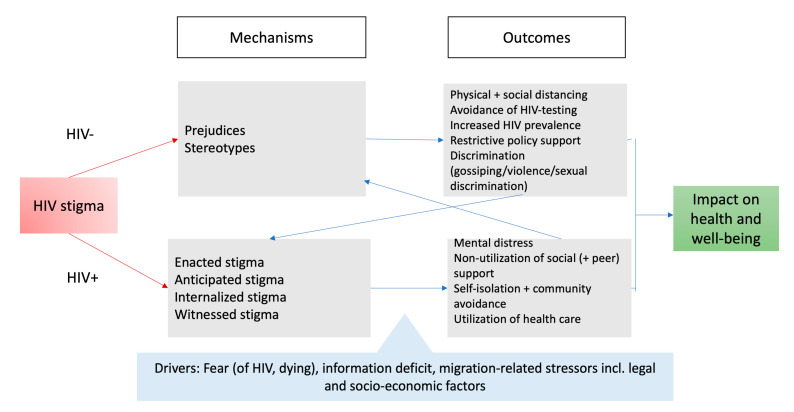
Contextualized HIV stigma framework for African communities in Belgium.

**Table 1 ijerph-18-08635-t001:** FGD and IDI participants’ socio-demographic characteristics.

FGD Participants, *n* = 76				IDI Participants, *n* = 20		
Characteristics	Female *n* (%)	Male *n* (%)	Total *n* (%)	Female *n* (%)	Male *n* (%)	Total *n* (%)
Total	37 (48.7)	39 (51.3)	76 (100)	10 (50)	10 (50)	20 (100)
Median age in years (min, max)	43 (18, 62)	37 (18, 64)	39 (18, 64)	39.5 (30, 58)	42 (29, 51)	41 (29, 58)
Median years of residence duration (min, max)	13 (2, 39)	14 (2, 28)	13 (2, 39)	6 (1, 25)	10 (2, 23)	6.5 (1, 25)
Marital status						
Married/cohabitant	17 (45.9)	16 (42.1)	33 (44.0)	2 (20.0)	1 (10.0)	3 (15.0)
Single/separated/divorced/widower	20 (54.1)	22 (57.9)	42 (56.0)	8 (80.0)	9 (90.0)	17 (85.0)
Religion						
Christian	33 (89.2)	28 (71.8)	61 (80.3)	9 (90.0)	9 (90.0)	18 (90.0)
Islam	3 (8.1)	6 (15.4)	9 (11.8)	1 (10.0)	1 (10.0)	2 (10.0)
Traditional		3 (7.7)	3 (3.9)			
None	1 (2.7)	2 (5.1)	3 (3.9)			
Source of income						
No income	10 (27.0)	8 (20.5)	18 (23.7)	1 (10.0)	5 (50.0)	6 (30.0)
Employed	14 (37.8)	14 (35.9)	28 (36.8)	3 (30.0)		3 (15.0)
Social welfare	7 (18.9)	7 (17.9)	14 (18.4)	5 (50.0)	5 (50.0)	10 (50.0)
Other	5 (13.5)	9 (23.1)	14 (18.4)			
Educational level						
None/primary school	1(2.7)	2 (5.1)	3 (3.9)	1 (10.0)		1 (5.0)
Vocational school/technical school	4 (5.4)	2 (5.1)	6 (7.9)	2 (20.0)	3 (30.0)	5 (25.0)
Secondary school	18 (48.6)	14 (35.9)	32 (42.1)	3 (30.0)	3 (30.0)	6 (30.0)
High school/university	14 (37.8)	21 (53.8)	35 (46.0)	4 (40.0)	4 (40.0)	8 (40.0)
No health insurance	6 (16.2)	6 (15.4)	12 (15.8)	2 (20.0)	4 (40.0)	6 (30.0)

**Table 2 ijerph-18-08635-t002:** IDI participants’ medical characteristics.

Characteristics	Female (%)	Male (%)	Total (%)
Total	10 (50.0)	10 (50.0)	20 (100.0)
Median years with HIV (min, max)	10.5 (2, 24)	12.5 (4, 23)	12.0 (2, 24)
Country of diagnosis			
Belgium	4 (40.0)	8 (80.0)	12 (60.0)
Country of origin	5 (50.0)	2 (20.0)	7 (35.0)
Elsewhere	1 (10.0)		1 (5.0)
Reasons of diagnosis			
Check-up	5 (50.0)	3 (30.0)	8 (40.0)
Sickness	3 (30.0)	6 (60.0)	7 (35.0)
Pregnancy	1 (10.0)		1 (5.0)
After death of a partner living with HIV	1 (10.0)		1 (5.0)
Seeking asylum		1 (10.0)	1 (5.0)
On Antiretroviral medication	10 (100.0)	10 (100.0)	20 (100.0)
Median years of duration on medication (min, max)	10.0 (1, 23)	10.5 (3, 23)	10.5 (1, 23)
Affiliation to a peer-support group	5 (50.0)	3 (33.33)	7 (35.0)

## Data Availability

The datasets generated for this study are available on request to the corresponding author.

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
