# Peer review of "Stigma Mechanisms and Outcomes among Sub-Saharan African Descendants in Belgium—Contextualizing the HIV Stigma Framework"

_ijerph, 2021, doi:10.3390/ijerph18168635_

Round 1
Reviewer 1 Report
Authors have collected a very rich and interesting material composed of 10 focal groups and 20 in deep interviews with PLHIV originating from SSA living in Belgium. As stated by authors, they used a theoretical framework primarily designed to understand and measure individual processes of stigmatization, aiming to explore HIV-related stigma and its underlying drivers and outcomes among studied communities.
There were relevant findings. Nevertheless, results were reported in a telegraphic and shallow manner that did not allow moving forward in understanding the processes that foster the intersecting of HIV- and migration-related stigmata, that in my opinion is central to study object.
One main finding was that gossiping was revealed as a significant factor in re-iterating existing stereotypes about PLHIV among study population. Gossip may be an important instrument for social control and a study about stigma should value this finding. How does gossip operates in the study population? What aspects in the migrant context are more affected by gossip? The paper would greatly improve relevance if authors discuss the role reported gossip in the study communities. How these findings impact the observed HIV-related stigma outcomes among immigrants?
Author Response
Responses of the authors to reviewers
Responses to Reviewer 1 Comments
Point 1: Results were reported in a telegraphic and shallow manner that did not allow moving forward in understanding the process that foster the intersecting of HIV- and migration-related stigmata (crucial)
Response 1:
We agree with the author’s remark about the relevance of the intersecting stigmata given the evidence in the literature (i.e body colour, racism, religion, gender, sexual orientation, immigration status, etc (Ziersch et al, 2021; Logie et al, 2013) and impact on the life of the immigrants.
However, our study didn’t specifically look for intersecting migration-related stigmata rather it limited its scope to what is happening in African communities. Nevertheless, we identified some issues that have been reported by participants but they were not deeply discussed. We improved the section 3.6.3. in this regard. Multiple forms of stigma and migration-related stressors (see line 476-497).
Moreover, we edited the manuscript for better fluidity of the text. We would like to point out that we made an effort to include a broad variety of themes that emerged from the data and describe them adequately. On the other hand, this means that the narratives have to be concise.
Point 2: One main finding was that gossiping was revealed as a significant factor in re-iterating existing stereotypes about PLHIV among study population. Gossip may be an important instrument for social control and a study about stigma should value this finding. How does gossip operates in the study population? What aspects in the migrant context are more affected by gossip? The paper would greatly improve relevance if authors discuss the role reported gossip in the study communities. How these findings impact the observed HIV-related stigma outcomes among immigrants?
Response 2.
We thank the referee for this relevant remark. We elaborated on how gossiping operates in the communities and the aspects that are affected by the gossip in this population. (see line 326-378). Also, we took into account the need of improving the discussion on the gossip (see line 589-602)
References:
Anna Ziersch , Moira Walsh , Melanie Baak , Georgia Rowley , Enaam Oudih and Lillian Mwanri. It is not an acceptable disease”: A qualitative study of HIV-related stigma and discrimination and impacts on health and wellbeing for people from ethnically diverse backgrounds in Australia. BMC Public Health (2021) 21:779 https://doi.org/10.1186/s12889-021-10679-y
Logie, C., James, L., Tharao, W., & Loutfy, M. (2013). Associations between HIV-related stigma, racial discrimination, gender discrimination, and depression among HIV-positive African, Caribbean, and Black women in Ontario, Canada. AIDS Patient Care and STDs, 27(2), 114–122.
Reviewer 2 Report
This is an interesting research, but it needs to clarify many methodological details so that we can actually analyze and understand the text. With this in mind, I raise the following questions to the authors:
1. "Although this framework was primarily designed to understand and measure individual processes of stigmatization, here we used it as guidance for exploring HIV-related stigma and its underlying drivers among communities and PLHIV originating from SSA living in Belgium" about "this adaptation" do the authors agree? Is there theoretical support for this?
2. This cross-sectional, multi-site qualitative study used an interpretive ethnographical approach - Question: Some of these terms are unusual and it would be interesting references capable of supporting it;
3. According to Lincon and Denzin (2006), qualitative research has been changing over time, with the 1990s being a milestone in this process. Points of these changes are the narrative turn marked by gender and the replacement of foundationalist epistemology by constructivist, hermeneutic, feminist, post-structural, pragmatist, critical theory of race and queer theory. This context of changes where "no one believes in the concept of a unified sexual subject anymore and not even in any unified subject" makes the researcher an "interpretive bricoleur" who knows how to interview, observe, study material culture, think within the limits of visual methods and exceed these limits; writing poetry, fiction and self-ethnography; build narratives that contain explanatory stories; use qualitative software; do text-based investigations, build testimonies using focus group interviews; and even engaging in applied ethnography and policy-making” (page 363-364). These changes, which strengthen and highlight the contribution of qualitative research, keep in focus the issue of validity and reliability, which is a constant challenge. How did the authors handle this in the manuscript? What are the validation characteristics of the instruments used to measure the object of study?
4. The definition of sampling is confusing. “Closing the sample means defining the set that will support the analysis and interpretation of data. In non-probabilistic (intentional) samples, such definition is made based on the researcher's experience in the field of research, in a decision based on reasoning guided by theoretical knowledge of the relationship between the object of study and the corpus/speech to be studied. If there was no closure due to exhaustion (approaching all eligible subjects), it must be justified why the processing of new observations and the recruitment of new participants were interrupted. One of the ways to do this corresponds to the theoretical saturation sampling process: data collection is interrupted when it is found that new elements to support the desired theorization (or possible under the circumstances) are no longer inferred from the field of observation ”. This process involves a coming and going between data collection and analysis, as it is necessary for the researcher to realize in the elaboration of analysis categories that the data does not imply the emergence of new categories or pre-categories. Therefore, it is not about the number of participants, because few qualitative research works with a number as large as the number of the study in question.
5. The analysis process and results should be described in minimum and sufficient detail so that readers have a clear understanding of how the analysis was carried out, its strengths and limitations.
6. How were the speeches selected? what process were they chosen by? were randomly allocated?
Author Response
Responses of the authors to reviewers
Responses to Reviewer 2 Comments
Table: Introduction: not sufficient background and relevant references
Response 1:
Considering the length of the results, we wanted to keep the introduction concise. However, we added some references related to the evidence on low HIV prevention demand in the study population. (see line 36-41)
Also, we deleted the HIV Stigma framework of Earnshaw and Chaudoir, because we could not pertain permission to use the original figure on time. However, the framework’s content has been sufficiently explained in the text.
Point 1: "Although this framework was primarily designed to understand and measure individual processes of stigmatization, here we used it as guidance for exploring HIV-related stigma and its underlying drivers among communities and PLHIV originating from SSA living in Belgium" about "this adaptation" do the authors agree? Is there theoretical support for this?
Do the authors agree about this adaptation?: Yes, we agree
Is there theoretical support for this ?
Response :
We based our research on the existing framework which has been constructed based on an extensive review (Earnshaw and Chaudoir, 2009). Our results presented similarities and some difference contextualizing it for our study setting. The qualitative approach provides an in-depth understanding of what the frameworks’ constructs mean in the given context.
Several authors have also adopted this framework using a qualitative research approach, e.g. Reinius M et al, 2021 & Fauk, N.K et al, 2021)
We also specify that additional research is needed to deepen and quantify our findings.
Point 2: This cross-sectional, multi-site qualitative study used an interpretive ethnographical approach - Question: Some of these terms are unusual and it would be interesting references capable of supporting it.
Response Point 2: We consider that the study design is clear enough, but we added the reference (Wilson, William Julius, and Anmol Chaddha. 2010).
Point 3:
… these changes, which strengthen and highlight the contribution of qualitative research keep in focus the issue of validity and reliability, which is a constant challenge. How did the authors handle this in the manuscript? What are the validation characteristics of the instruments used to measure the object of study?
Response 3: Validity results was assured by different types of triangulation: inclusion of participants from different background e.g age, gender, education level, origin, length of living with or without HIV in Europe; different data sources (FGD and IDIs) and the use of different researchers in anonymizing and interpreting the data. In addition, feedback of communities was obtained presenting the research results.
This information has been integrated in the section “Strengths and Limitations (see line 592-595)
- The definition of sampling is confusing. “Closing the sample means defining the set that will support the analysis and interpretation of data. In non-probabilistic (intentional) samples, such definition is made based on the researcher's experience in the field of research, in a decision based on reasoning guided by theoretical knowledge of the relationship between the object of study and the corpus/speech to be studied. If there was no closure due to exhaustion (approaching all eligible subjects), it must be justified why the processing of new observations and the recruitment of new participants were interrupted. One of the ways to do this corresponds to the theoretical saturation sampling process: data collection is interrupted when it is found that new elements to support the desired theorization (or possible under the circumstances) are no longer inferred from the field of observation”. This process involves a coming and going between data collection and analysis, as it is necessary for the researcher to realize in the elaboration of analysis categories that the data does not imply the emergence of new categories or pre-categories. Therefore, it is not about the number of participants, because few qualitative research works with a number as large as the number of the study in question.
Response 4: We agree that the qualitative research can include different sample sizes resulting in low or large numbers of the study participant. We took an iterative approach and didn’t determine the number of participants beforehand. Data saturation was reached at the point where we had included “10 FGD with 76 participants and 20 PLHIV” because no additional input was obtained from the participants at that stage. That is what we refer to in the phrase in the manuscript when we say “ thematic saturation was achieved”.
We restructured the sections “sampling and recruitment.” (see line 91-129)
- The analysis process and results should be described in minimum and sufficient detail so that readers have a clear understanding of how the analysis was carried out, its strengths and limitations.
Response 5: We described in detail the analysis steps (see line 149-181).
- How were the speeches selected? What process were they chosen by? Were they randomly allocated?
Response 6: Quotes were selected based on rich data and most illustrative ones for the themes under description, being representative of the data collected and succinct (Lorelei Lingard, 2019). Thanks to the remark, we could review quotes and made them more representative of the data.
References
Lorelei Lingard. Beyond the default colon: Effective use of quotes in qualitative research Perspect Med Educ (2019) 8:360–364 https://doi.org/10.1007/s40037-019-00550-7
Round 2
Reviewer 1 Report
Authors have reviwed the paper properly.
Author Response
Thank you very much for reviewing.
Reviewer 2 Report
Unfortunately, I don't believe the authors' responses were satisfactory enough that I nominated the article for publication. The manuscript has many validity problems that do not support it. I'm going to name just 3:
1. The framework adaptation - I didn't ask if you authors agreed with this adaptation, but if the original proposal focused on stigmatization agreed with the change or not. You cannot simply change this without the consent of the original authors, much less apply it to someone completely different from what it was proposed;
2. Sampling and data saturation criteria: The methods adopted by the authors are not aligned with the canons and principles of qualitative research, not even with the check-lists (COREQ) indicated by the Equator network and followed by major journals such as the Lancet. It is not possible to define saturation criteria while the research is taking place as the authors refer; Much less, because no new entrants can be obtained. This is simply wrong and I suggest the following readings:
Graneheim U.H., Lundman B. Qualitative content analysis in nursing research: concepts, procedures and measures to achieve trustworthiness. Nurse Education Today, 2004. V.24, p. 105–112. doi:10.1016/j.nedt.2003.10.001
Elo S., Kynga's H. The qualitative content analysis process. J Adv Nurs. 2008. 62(1):107-15. doi: 10.1111/j.1365-2648.2007.04569.x.
Fontanella B.J.B. et al. Sampling in qualitative research: proposal of procedures to verify theoretical saturation. Public Health Cad, 2011. V.27(2):389-394. doi: 10.1590/S0102-311X2011000200020
Fontanella BJB, Ricas J, Turato ER. Saturation sampling in qualitative health research: theoretical contributions. CAd Public Health 2008; 24:17-27.
Denzin N.K. and Lincoln Y. Qualitative research planning - Theory and Approaches. 2006. Publisher ArtMed.
3. Selecting testimonials according to their "rich data", is biasing the data. The data are left with a kind of "selection bias" for the researcher, who can choose what he/she thinks is appropriate and not what the participants brought as the most relevant.
Therefore, I believe that the manuscript cannot be accepted for this prestigious Journal, as it has no internal and external validity.
Author Response
Dear
Please see the attachment
